# Designing isolation guidelines for COVID-19 patients with rapid antigen tests

Yong Dam Jeong [1,2,13], Keisuke Ejima [3,4,13] ✉, Kwang Su Kim[1,5,13], Woo Joohyeon[1], Shoya Iwanami [1], Yasuhisa Fujita[1], Il Hyo Jung[2], Kazuyuki Aihara[6], Kenji Shibuya [4], Shingo Iwami [1,7,8,9,10,11] ✉, Ana I. Bento [3] & Marco Ajelli[12]

Appropriate isolation guidelines for COVID-19 patients are warranted. Currently, isolating for fixed time is adopted in most countries. However, given the variability in viral dynamics between patients, some patients may no longer be infectious by the end of isolation, whereas others may still be infectious. Utilizing viral test results to determine isolation length would minimize both the risk of prematurely ending isolation of infectious patients and the unnecessary individual burden of redundant isolation of noninfectious patients. In this study, we develop a data-driven computational framework to compute the population-level risk and the burden of different isolation guidelines with rapid antigen tests (i.e., lateral flow tests). Here, we show that when the detection limit is higher than the infectiousness threshold values, additional consecutive negative results are needed to ascertain infectiousness status. Further, rapid antigen tests should be designed to have lower detection limits than infectiousness threshold values to minimize the length of prolonged isolation.

Vaccination campaigns for COVID-19 are being successfully implemented over the world[1]. However, despite the high vaccination coverages achieved in many Western countries[1], the emergence of the Omicron variant reminded us how the vaccination alone may not be sufficient to prevent new major waves of infection[1]. Non-pharmaceutical interventions (NPIs), such as wearing masks, social distancing, reactive closures, still play a central role in the pandemic response, and testing, isolation, and quarantine represent its backbone[2].

The discussion regarding the isolation of SARS-CoV-2 infected individuals and when to end their isolation period remains contentious. A longer isolation decreases the risk of transmission; however, it may impose unnecessarily lengthy isolation, which poses both a burden on physical and mental health of the patients[3] and in the economy at large[4]. Thus, creating flexible, evidence-based criteria for determining appropriate length of patient's isolation is paramount.

There are two main approaches widely adopted by countries to determine the end of the isolation of COVID-19 patients. One is to

---

[1]interdisciplinary Biology Laboratory (iBLab), Division of Biological Science, Graduate School of Science, Nagoya University, Nagoya, Japan. [2]Department of Mathematics, Pusan National University, Busan, South Korea. [3]Department of Epidemiology and Biostatistics, Indiana University School of Public Health-Bloomington, Bloomington, IN, USA. [4]The Tokyo Foundation for Policy Research, Tokyo, Japan. [5]Department of Scientific computing, Pukyong National University, Busan, South Korea. [6]International Research Center for Neurointelligence, The University of Tokyo, Tokyo, Japan. [7]Institute of Mathematics for Industry, Kyushu University, Fukuoka, Japan. [8]Institute for the Advanced Study of Human Biology (ASHBi), Kyoto University, Kyoto, Japan. [9]NEXT-Ganken Program, Japanese Foundation for Cancer Research (JFCR), Tokyo, Japan. [10]Interdisciplinary Theoretical and Mathematical Sciences Program (iTHEMS), RIKEN Saitama, Japan. [11]Science Groove Inc, Fukuoka, Japan. [12]Laboratory for Computational Epidemiology and Public Health, Department of Epidemiology and Biostatistics, Indiana University School of Public Health-Bloomington, Bloomington, IN, USA. [13]These authors contributed equally: Yong Dam Jeong, Keisuke Ejima, Kwang Su Kim. ✉e-mail: kejima@iu.edu; iwami.iblab@bio.nagoya-u.ac.jp

isolate infected patients over fixed time, whereas the other is to isolate infected patients until their viral load drops below a threshold value[5]. In our previous study, we demonstrated that the latter approach, based on PCR testing of isolated individuals, could minimize unnecessary isolation while controlling the risk of further transmission[6]. This is because some patients are no longer infectious by the end of isolation (thus they are redundantly isolated), whereas others may still be infectious, due to substantial individual variability in viral dynamics[7]. However, PCR tests have some limitations when used to determine the end of isolation. First, the turnaround time is a day or two[8], suggesting that patients need to wait a day or two until they are released from isolation even though they were not infectious anymore. Second, PCR tests are expensive. For example, in the US, the cost of single PCR test is approximately 51 USD[9], whereas that of rapid antigen tests (i.e., lateral flow tests) is 5 USD[10], although the cost could differ between countries. Further, the facilities for PCR tests are not available everywhere.

Recently, as the third year of the pandemic as rolled in, the Centers for Disease Control and Prevention (CDC) created US based guidelines for when to relax precautions (thus isolation) for COVID-19 patients in health care settings[5]. In the early phase of the pandemic, the guideline included the use of PCR tests as follows: "Results are negative from at least two consecutive respiratory specimens collected ≥24 hours apart" (a test-based guideline)[5]. However, on August 10, 2020, possibly due to the discussed limitations of PCR testing, the guideline was updated as follows: "At least 10 days have passed since symptoms first appeared", because "in the majority of cases, it [a test-based guideline] results in prolonged isolation of patients who continue to shed detectable SARS-CoV-2 RNA but are no longer infectious[5]".

Given these limitations of PCR tests, the advantages of using antigen tests (i.e., lateral flow tests) to determine the end of the isolation period should be considered. Antigen tests have (i) shorter turnaround time (less than an hour)[8,11–13]; (ii) low cost, and (iii) easier accessibility when compared with PCR tests. Nevertheless, the low sensitivity of rapid antigen tests could be an issue. The detection limit of antigen tests is about $10^{5.0}$ copies/mL[11–14], whereas that of PCR tests is about $10^{2.0}$ copies/mL[15–17]. Further, the infectiousness threshold values assessed by epidemiological data and in-vivo experiments (i.e., culturability) were estimated to be $10^{5.0-6.0}$ [18,19], which is close to or slightly higher than the detection limits of antigen tests. This supported the use of antigen test screening to mitigate transmission risk[8,20,21]. We stress that our analyses use viral load rather than Ct value. Therefore, the detection limit represents the lowest value of the viral load that an antigen test can detect. Similarly, the infectiousness threshold represents the lowest value of the viral load for which SARS-CoV-2 transmission may occur.

Here, we use a mathematical modeling framework to study and evaluate the use of antigen tests in determining the optimal end of the isolation period. We submit that this will contribute to minimizing both the risk of onward transmission following isolation and the burden of the isolation.

## Results
### Descriptive statistics
In total, 10 papers included at least one patient meeting the inclusion criteria, and 210 cases were identified. Among those 210 cases, 109 and 101 cases were symptomatic and asymptomatic, and 85, 117, and 8 cases were reported from Asia, USA, and Europe, respectively (Table 1). In most studies, cycle thresholds were reported instead of viral load. Therefore, the cycle threshold was converted to viral load (copies/mL) using the conversion formula: $\log_{10}$(Viral load[copies/mL]) = $-0.32 \times$ Ct values[cycles] $+ 14.11$[22]. All the patients included in these studies were hospitalized regardless of their symptom status; however, clinical courses of infection (i.e., severity) were not consistently available.

## Model fitting to the symptomatic and asymptomatic individuals
Figure 1 shows the fitted curves of viral load for symptomatic and asymptomatic patients using estimated fixed effect parameters. For both cases, the peak viral load appears about 4 days after infection. However, the peak viral load was higher in symptomatic cases (about $10^{6.5}$ copies/mL for symptomatic cases vs. $10^{6.0}$ copies/mL for asymptomatic cases), and the viral load remained relatively high for longer time in symptomatic individuals. The viral load drops below 1 copy/mL at day 25 (95%PrI: 21–29) and 21 (95%PrI: 17–24) for symptomatic and asymptomatic cases, respectively. The difference on peak values of the viral load between symptomatic and asymptomatic cases was observed, which is explained by difference on the rate constant for virus infection in the model (Supplementary Table 1). The quicker clearance of the virus in asymptomatic individuals is explained by stronger immune response, with a higher death rate of infected cells in the model (Supplementary Table 1). This finding is in agreement with previous studies suggesting lower viral load and shorter persistence of viral RNA in mild than in severe cases[23–25] and longer persistence of viral RNA in symptomatic individuals[26]. The posterior distributions of the parameters are available in Supplementary Fig. 1. We further run the same analysis with different conversion formulas. However, substantial difference in the dynamics was observed (Supplementary Fig. 2). Given these differences in the viral dynamics, we evaluate different isolation guidelines for symptomatic and asymptomatic individuals.

## Antigen tests to end isolation
Figures 2 and 3 show the probability of prematurely ending isolation (risk) and the length of unnecessarily prolonged isolation (burden) for symptomatic and asymptomatic cases, respectively, by varying the consecutive negative results, intervals between tests, and infectiousness threshold values. The detection limit of rapid antigen tests was assumed to be $10^4$ copies/mL and $10^6$ copies/mL in Figs. 2 and 3, respectively. As we observed in our previous study[6], regardless of detection limits, infectiousness threshold values, and symptom presence, the risk declined as the interval between tests becomes longer and more consecutive negative results are needed. Meanwhile, the burden increased at the same time. Supplementary Fig. 3 shows the distribution of the risk and burden for representative scenarios (infectiousness threshold: $10^{5.0}$ copies/mL, detection limit: $10^{4.0}$ copies/mL [corresponding to Fig. 2] or $10^{6.0}$ copies/mL [corresponding to Fig. 3], consecutive negative results: twice, interval between tests: 1 to 5 days).

Should 5% or lower risk of prematurely ending isolation be considered as acceptable, it is not possible to identify a single optimal strategy as the effectiveness of the guideline is estimated to depend on the infectiousness threshold, detection limits of the antigen test, and symptom presence. For example, when the detection limit and infectiousness threshold value were $10^4$ copies/mL and $10^5$ copies/mL, the optimal guideline (denoted by the squares in Fig. 2b) for symptomatic individuals was to perform tests every day and to observe 2 consecutive negative results before ending the isolation (risk: 2.6% [95% prediction interval: 1.8% to 3.4%] and burden: 3.9 days [95%PrI: 0 to 10]). Supplementary Fig. 3 presents the distributions of the risk and burden of representative scenarios. The optimal guideline also depends on the acceptable risk of prematurely ending isolation. When a 1% or lower risk is considered to be acceptable, more consecutive negative results would be needed to end isolation. When the detection limit is high ($10^6$ copies/mL), an optimal guideline would require more consecutive negative results, as the infectiousness threshold values are below the detection limit and a limited number of consecutive negative results cannot guarantee that the viral load is below the infectiousness threshold.

Figure 4 summarized the burden of isolation when considering the identified optimal guideline under different conditions (i.e.,

**Table 1 | Summary of the viral load data used for modeling**

| Country | Number of data | Reporting unit | Specimens for measuring viral load | Date of collection | Source |
|---|---|---|---|---|---|
| **Symptomatic** | | | | | |
| USA | 33 | viral load (copies/mL) | Nares and oropharyngeal swabs | Nov 2020 to May 2021 | ref. 44 |
| USA | 12 | viral load (copies/mL) | Nares and oropharyngeal swabs | Nov 2020 to May 2021 | ref. 26 |
| Germany | 8 | viral load (copies/swab)[b] | Pharyngeal swab | Jan 2020 | ref. 19 |
| Korea | 34 | cycle threshold[a] | Oro/nasopharyngeal swabs | May 2020 | ref. 45 |
| Korea | 2 | cycle threshold[a] | Oro/nasopharyngeal swab | Feb 2020 | ref. 46 |
| Singapore | 12 | cycle threshold[a] | Nasopharyngeal swab | Jan to Feb 2020 | ref. 47 |
| China | 8 | cycle threshold[a] | Nasal swab | Jan 2020 | ref. 48 |
| **Asymptomatic** | | | | | |
| USA | 44 | viral load (copies/mL) | Nares and oropharyngeal swabs | Nov 2020 to May 2021 | ref. 44 |
| USA | 28 | viral load (copies/mL) | Nares and oropharyngeal swab | Nov 2020 to May 2021 | ref. 26 |
| Japan | 18 | cycle threshold[a] | Nasopharyngeal or throat swab | Jan 2020 | ref. 49 |
| Korea | 4 | cycle threshold[a] | Nasal and throat swabs | Feb to Apr 2020 | ref. 50 |
| Singapore | 7 | cycle threshold[a] | Nasopharyngeal swab | Mar to Apr 2020 | ref. 51 |

[a]Viral load was calculated from cycle threshold values using the conversion formula: $\log_{10}\left(\text{Viral load [copies/mL]}\right) = -0.32 \times \text{Ct values [cycles]} + 14.11$[22].

[b]1 swab = 3 mL.

## Symptomatic

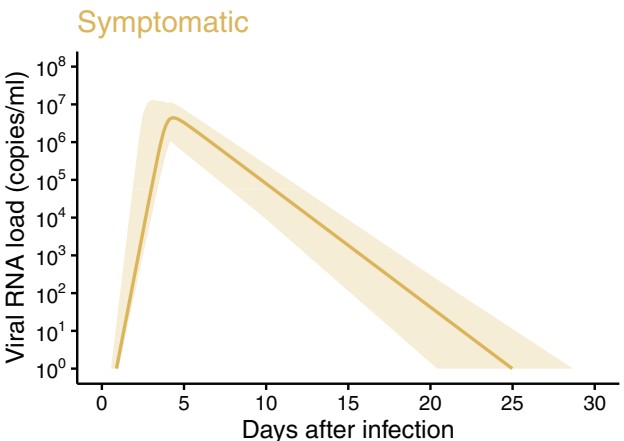

## Asymptomatic

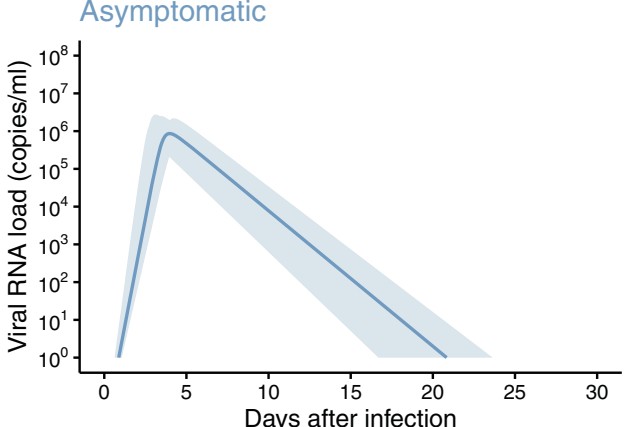

**Fig. 1 | Estimated viral load curves from the models for symptomatic and asymptomatic cases.** The solid lines are the estimated viral load curves for the best fit parameters of fixed effect. The shaded regions correspond to 95% prediction intervals. The 95% prediction intervals were created using bootstrap approach.

symptom presence, acceptable levels of risk, and infectiousness threshold values). Low burden was realized when higher risk could be accepted (comparison between Fig. 4a, b). The influence of symptom presence on the burden was estimated to be limited.

We performed several sensitivity analyses. First, we followed the approach of Han et al.[27]. and considered two alternative equations to convert Ct values to viral load. Although the obtained dynamics were slightly different depending on the selected conversion formula (Supplementary Fig. 2), the overall impact on our main results was not substantial (Supplementary Fig. 4).

Second, we set the initial day of the testing to be 5 days after the infection event (as compared to 8 days used in the main analysis). The obtained results were very consistent with those obtained in the main analysis (see Supplementary Fig. 5 as compared to Fig. 2). In fact, the viral load was much higher than the detection limit both at 5 and 8 days after infection; thus the false-negative rate was very low at both times (Fig. 1).

Third, we considered a fixed error of the viral load for each patient over the course of infection. The overall results were in agreement with the main analysis; however, we estimated a general increased risk of prematurely ending isolation, which is associated with patients with

consistently under-measured VL thus resulting in an earlier release (see Supplementary Fig. 6 as compared to Fig. 2).

The proportion of infectious patients (i.e., risk) does not capture how long prematurely released individuals are still infectious. To capture it, we have added a new metric: the mean number of days an infected individual is infectious after the end of the isolation. The obtained results are reported in Supplementary Fig. 7. For example, when an infectiousness threshold value of $10^{5.0}$ copies/mL, detection limit $10^{4.0}$ copies/mL, tests performed every day, and 2 consecutive negative results to end isolation of symptomatic individuals, the mean number of days was 2.1 days (95%PrI: 1 to 6).

The influence of the combination of infectiousness threshold values and detection limits on the burden was intriguing. When the detection limit was higher than the infectiousness threshold values (i.e., detection limit was $10^6$ copies/mL), the burden was minimized when the detection limit is close to the infectiousness threshold values. However, when the detection limit is lower than the infectiousness threshold values, the burden was not much influenced by the infectiousness threshold values. That says, rapid antigen tests should have lower detection limits than infectiousness threshold values, and the burden becomes large when the detection limit is much higher than

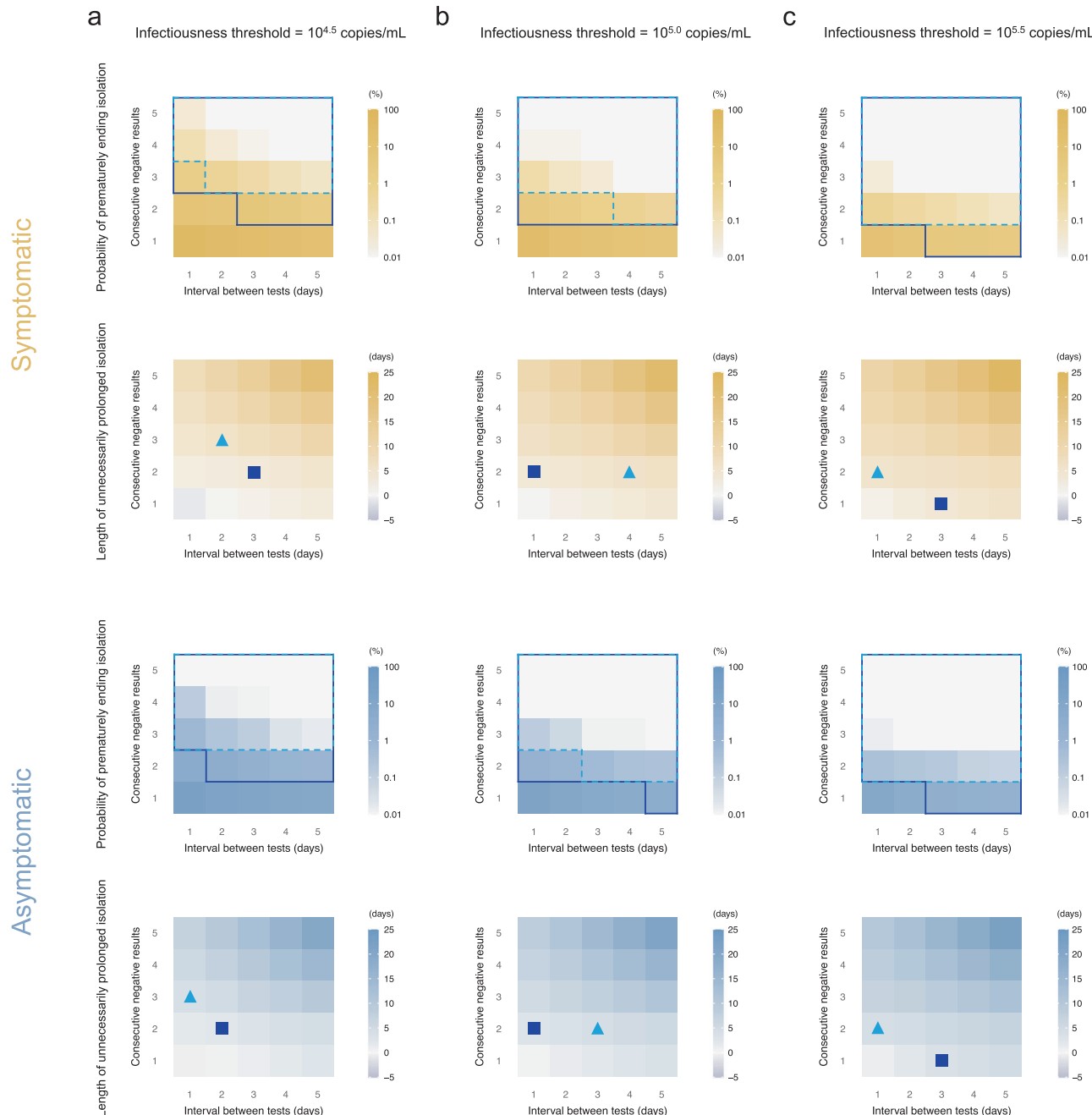

**Fig. 2 | Optimal isolation guideline for symptomatic and asymptomatic cases using antigen test (detection limit=$10^4$ copies/mL). a** Probability of prematurely ending isolation (upper panels) and mean length of unnecessarily prolonged isolation (lower panels) for different values of the interval between antigen tests and the number of consecutive negative results necessary to end isolation for each case; the infectiousness threshold value is set to $10^{4.5}$ copies/mL. The areas surrounded by sky-blue dotted lines and blue solid lines are those with 1% or 5% or lower of risk of prematurely ending isolation of infectious patients, respectively, and the triangles and squares correspond to the conditions which realize the shortest prolonged isolation within each area. **b** The same as **a**, but for an infectiousness threshold value of $10^{5.0}$ copies/mL. **c** The same as **a**, but for an infectiousness threshold value of $10^{5.5}$ copies/mL. Color keys and symbols apply to all panels. Note that the estimate values are based on 100 simulations with 1000 patients each for symptomatic and asymptomatic cases, respectively. Accordingly, 1000 parameter sets were sampled from the posterior distribution of each model parameter.

the infectiousness threshold values, even though the guidelines are optimized for given conditions.

## Discussion

We provide a data-driven quantitative assessment of alternative guidelines for the definition of optimal duration of the isolation period, based on the use of rapid antigen tests. We found that the optimal guideline was dependent on the acceptable risk, detection limits, infectiousness threshold values, in agreement with what was estimated for PCR-based exit testing guidelines[6]. Among those three factors, the detection limit was positively associated with consecutive negative results necessary to end isolation. In other words, more consecutive negative results are necessary when the detection limit is above infectiousness threshold values. Our study supports the need to define different testing strategies to end the isolation for symptomatic and asymptomatic individuals. Comparing the burden

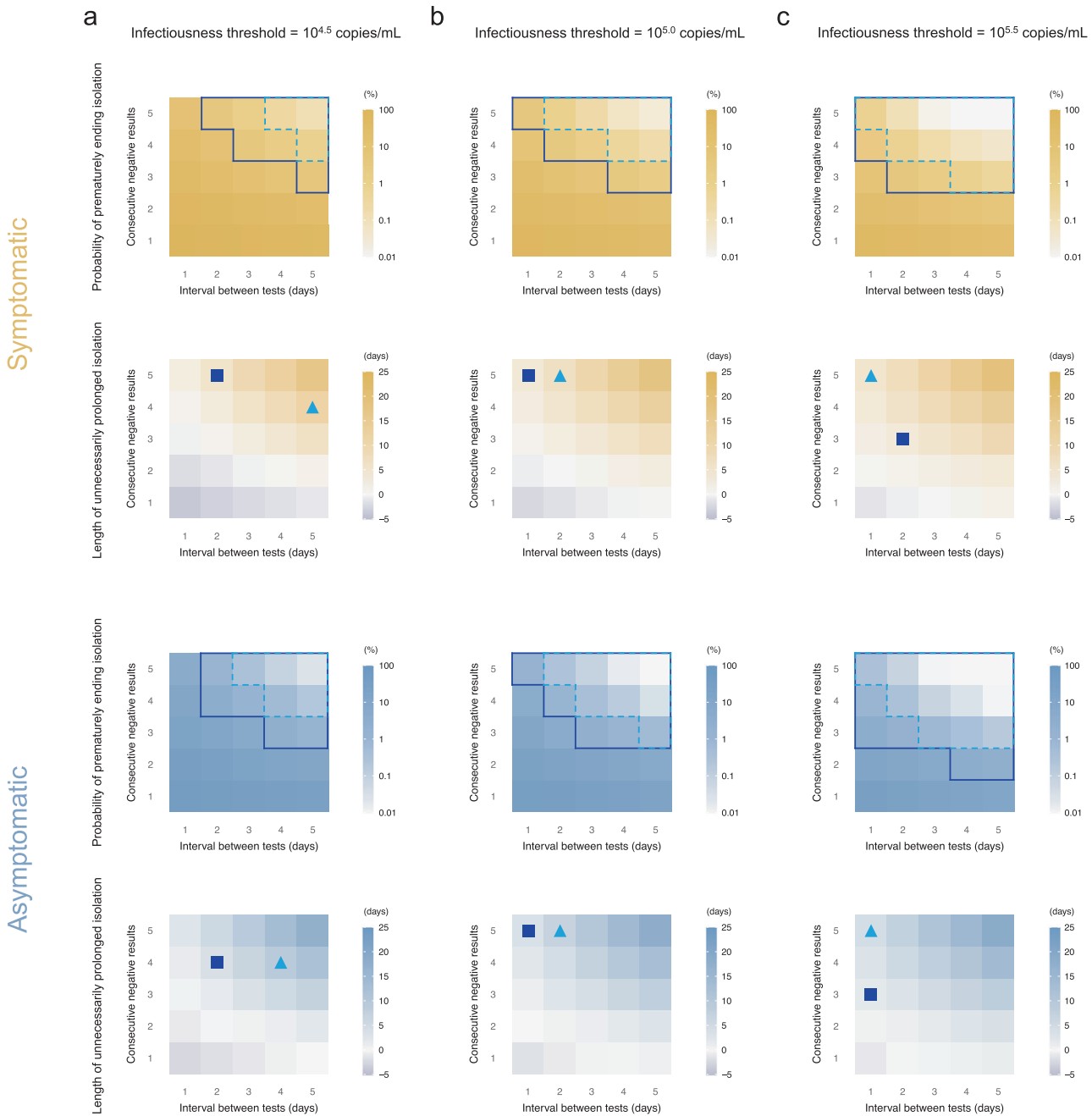

**Fig. 3 | Optimal isolation guideline for symptomatic and asymptomatic cases using antigen test (detection limit=$10^6$ copies/mL). a** Probability of prematurely ending isolation (upper panels) and mean length of unnecessarily prolonged isolation (lower panels) for different values of the interval between antigen tests and the number of consecutive negative results necessary to end isolation for each case; the infectiousness threshold value is set to $10^{4.5}$ copies/mL. The areas surrounded by sky-blue dotted lines and blue solid lines are those with 1% or 5% or lower of risk of prematurely ending isolation of infectious patients, respectively, and the triangles and squares correspond to the conditions which realize the shortest prolonged isolation within each area. **b** The same as **a**, but for an infectiousness threshold value of $10^{5.0}$ copies/mL. **c** The same as **a**, but for an infectiousness threshold value of $10^{5.5}$ copies/mL. Color keys and symbols apply to all panels. Note that the estimate values are based on 100 simulations with 1000 patients each for symptomatic and asymptomatic cases, respectively. Accordingly, 1000 parameter sets were sampled from the posterior distribution of each model parameter.

of isolation (i.e., length of prolonged isolation) depending on different settings, we found that rapid antigen tests should have lower detection limits than infectiousness threshold values, and the burden increases as the detection limit is much higher than the infectiousness threshold values, even though the guidelines are optimized for given conditions.

The burden of isolation under optimal guidelines was influenced by infectiousness threshold values, which was not observed in the previous study using PCR tests[6]. PCR tests can quantitatively measure viral load; thus, the measured viral load is directly compared against the infectiousness threshold value whatever the value is. Therefore, the impact of infectiousness threshold values was not observed on the burden of isolation under optimal guidelines when PCR tests are used[6]. Meanwhile, as results from rapid antigen tests are qualitative (i.e., positive, or negative), we only know whether the viral load is below the detection limit. However, we do not necessarily know whether it is below the infectiousness threshold value. For instance, if the detection limit is below the infectiousness threshold value (detection limit is $10^4$

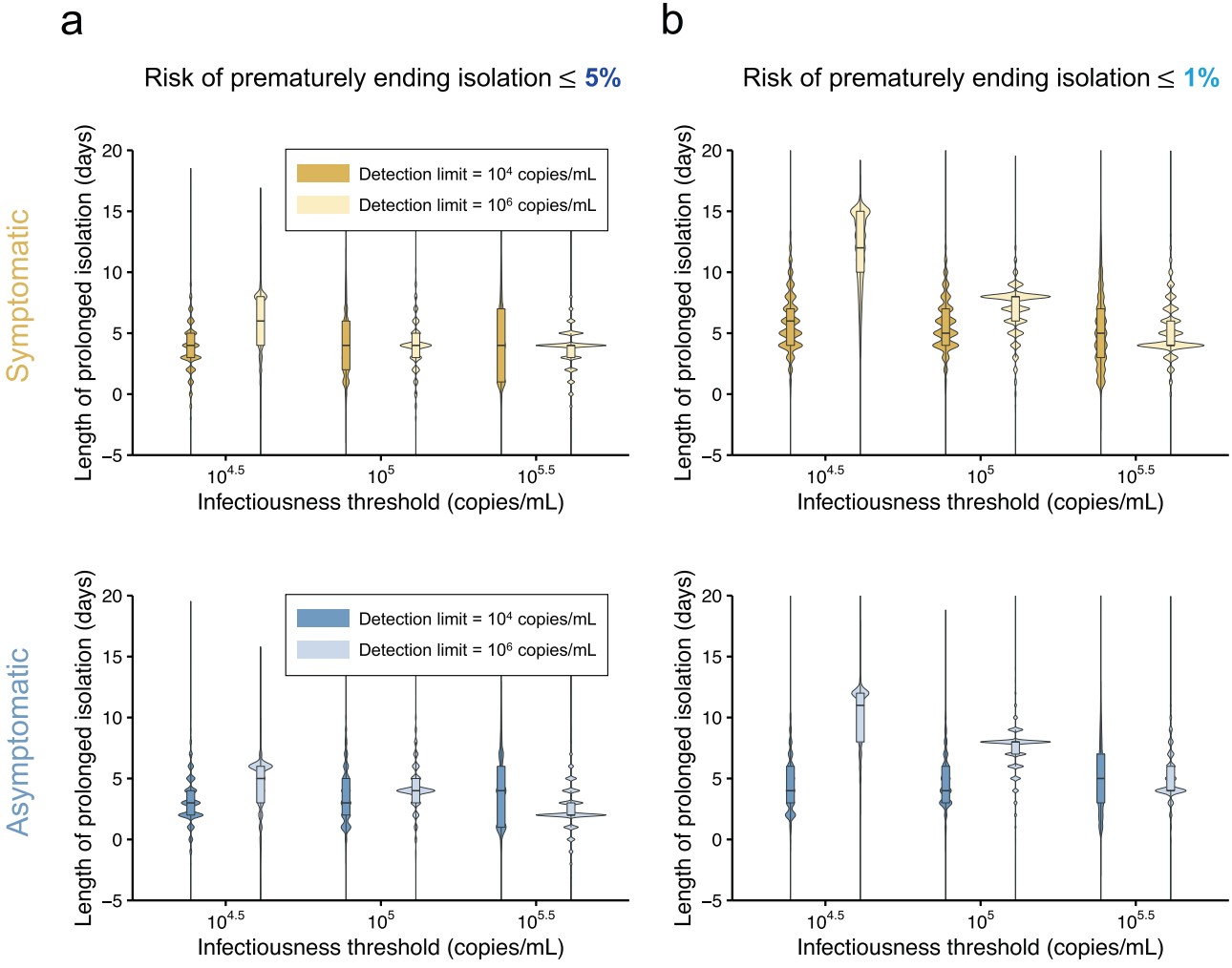

**Fig. 4 | Comparison between the situations of high and low detection limits for symptomatic and asymptomatic cases. a** Distributions of length of prolonged isolation for different infectiousness threshold values, detection limits, and symptom presence when considering a 5% or lower risk of prematurely ending isolation. The violin plots show the kernel probability density, whereas the box plots show the median (50 percentile; bold lines) and interquartile ranges (25 and 75 percentiles; boxes). Note that the interval between antigen tests and the number

of consecutive negative results necessary to end isolation were selected to minimize the duration of prolonged isolation. **b** The same as **a**, but considering a 1% or lower risk of prematurely ending isolation. Note that the estimate values are based on 100 simulations with 1000 patients each for symptomatic and asymptomatic cases, respectively. Accordingly, 1000 parameter sets were sampled from the posterior distribution of each model parameter.

copies/mL in this study), negative antigen tests results suggest that the viral load is below the infectiousness threshold value. In such case, we did not find much influence of infectiousness threshold values on the burden of isolation. Meanwhile, if the detection limit is above the infectiousness threshold value (detection limit is $10^6$ copies/mL in this study), negative antigen results do not necessarily suggest that the viral load is below the infectiousness threshold value. Therefore, in such cases, the burden increases when the difference between the infectiousness threshold value and the detection limit is large.

Our study highlights differences in the viral dynamics of symptomatic and asymptomatic infections. Specifically, we found that the viral load decays quicker in asymptomatic patients than symptomatic patients (although the 95%PrIs overlap); thus, the viral load of asymptomatic individuals fluctuates around the detection limit for shorter period, which leads to lower risk and less burden if the same guideline is applied to both symptomatic and asymptomatic patients. As infection from different variants appears to be associated with different severity rates (including probability of developing any symptom)[28–30], if the same guideline to end the isolation period is applied both to symptomatic and asymptomatic infections, the overall effectiveness of that guideline can vary for different SARS-CoV-2 variants.

Other studies have assessed the impact of using antigen tests in the context of ending isolation[20,31–33]. All these studies consistently concluded that using antigen test may reduce redundant isolations or prevent onward transmission. Similar to what we have done, Peng et al.[31] and Quilty et al.[20] considered the temporal change in the viral load, which in turns affect the transmission potential and test sensitivity. However, differently from those studies (which are based on piece-wise models or cubic Hermite splines), here we used a model that provides a biological explanation of the dynamics of viral load and can thus be refined to consider other factors shaping viral dynamics as, for instance, the use of an antiviral treatment[34].

While our results are robust, we would like to point out a few limitations. First, the data used to calibrate the model refer to the original SARS-CoV-2 lineage, presenting a limitation to our findings. Previous studies suggest the viral dynamics to be different between the original and the Delta variant[35]. Contrary, the generation time distributions of Alpha and Delta variants were suggested to be similar[36]. Note that more people are vaccinated or previously infected, which hinders pure comparison between variants. Second, as of December 2021 (the date we conducted our literature review), there were no publicly available data to calibrate the model for vaccinated

individuals, regardless of vaccine types and numbers of doses. Indeed, previous studies have shown difference in the viral load of infected vaccinated vs. infected unvaccinated individuals[37]. Third, we applied the same formula to convert Ct values to viral load despite this conversion should be determined for each study individually by considering the specific PCR assays[27] that were used. However, such information was not available to us to carry out better conversions. To mitigate this limitation, we considered alternative equations to convert Ct values to viral loads and found consistent results.

The COVID-19 pandemic is having an unprecedented impact on the lives of nearly every human being on this planet and is still causing interruptions in educational and economic activities. Isolating infected individuals is still a key component of the pandemic response and development of appropriate isolation guidelines is needed. Our study provides insights on the use of rapid antigen tests to minimize both the burden of isolation and the risk of releasing infectious individuals, and suggests that different guidelines may be warranted for symptomatic and asymptomatic individuals.

## Methods
### Viral load data
Longitudinal viral load data of symptomatic and asymptomatic COVID-19 patients were extracted through PubMed search and regular monitoring of COVID-19 literature. Specifically, we used the following query in PubMed:

("COVID-19" or "SARS-CoV-2") and ("viral load" or "viral loads" or "viral titer" or "cycle threshold" or "cycle thresholds" or "viral RNA concentration" or "viral RNA concentrations" or "viral shedding") and ("peak" or "kinetics" or "clinical course")

A total of 250 papers published in 2020 and 2021 were included for further investigation. We reviewed each paper to extract the relevant data based on the following inclusion criteria: 1) viral load was measured at least at three different time points; 2) viral load was measured from upper respiratory specimens (i.e., nose or pharynx); 3) patients were not treated with antiviral drugs or vaccinated before infection (as our model accounts neither for vaccination nor antiviral treatment). A total of 7 papers met these criteria. We have further identified 3 studies by our regular search of the scientific literature through PubMed and Google Scholar. All data refer to alpha, epsilon, and non-variants of interest/variants of concern (VOI/VOCs) as well as the ancestral lineage. All the data used in this study comes from published de-identified data, ethics approval was not needed.

### Modeling SARS-CoV-2 viral dynamics and parameter estimation
The viral load data were used to parameterize the mathematical model of viral dynamics, which describes temporal change in viral load over the course of SARS-CoV-2 infection in each infected individual. The model was previously proposed for infectious diseases causing acute infection and utilized in SARS-CoV-2 research[6,7,34,38,39]:

$$\frac{df(t)}{dt} = -\beta f(t) V(t), \tag{1}$$

$$\frac{dV(t)}{dt} = \gamma f(t)V(t) - \delta V(t). \tag{2}$$

The first variable $f(t)$ is the ratio between two numbers: the number of uninfected target cells at time $t$ and the number of uninfected target cells at the time of infection ($t = 0$). The second variable $V(t)$ is the amount of virus per unit of sample specimens (copies/mL) at time $t$. The three parameters in the model $\beta$, $\gamma$, and $\delta$ are the rate constant for virus infection, the maximum viral replication rate (when infected cells are limited), and the death rate of infected cells, respectively. The time origin of the longitudinal viral load data corresponds to the time after symptom onset (for symptomatic patients) and the time

after diagnosis (for asymptomatic patients). Therefore, we estimated a further model parameter, $\tau$, which represents the time interval between infection to symptom onset for symptomatic patients or to diagnosis for asymptomatic patients (see Supplementary Note 1 for detail). Note that we used day as a unit of time in this study, and time 0 is the time of infection. Therefore $V(0) = 10^{-2}$ (copies/mL) and $f(0) = 1$, following the previous study[40]. Under reasonable parameter setting, the trajectory of viral load $V(t)$ shows a bell-shaped curve; the viral load increases exponentially first, hits the peak, and then declines because of limited uninfected target cells that monotonically decrease as virus increases.

A nonlinear mixed-effect model was used for parameter estimation[6]. The nonlinear mixed-effect model assumes both fixed effect and random effect, where fixed effect captures the viral dynamics which are common in the population, whereas the random effect captures the difference between individuals[41,42]. The time of infection for each individual was estimated through the fitting. Model parameters were estimated independently for symptomatic and asymptomatic patients. For more detail of fitting, see Supplementary Note 1.

### Simulation of viral dynamics and ending isolation following different guidelines
The "true" viral load data, $V(t)$, for 1000 simulated patients was estimated by running the developed viral dynamics model. Parameter values of the simulation for each patient were sampled from the joint posterior distributions of model parameters (as estimated in the fitting process Supplementary Fig. 8). The measured viral load is assumed as a sum of the true viral load and the error: $\hat{V}(t) = V(t) + \varepsilon, \varepsilon \sim N(0, \sigma)$, where $\varepsilon$ is the error term, which is defined as the difference between the viral load reported in the data and the viral load estimated by the calibrated model. The variance of the error term, $\sigma^2$, was estimated in the fitting process (see Supplementary Note 1).

We assumed that the isolation and the first test were performed 8 days after infection. The test is repeated with a fixed time interval until a fixed number of consecutive negative results ($\hat{V}(t) <$ detection limit) are observed. To simulate different guidelines, we varied the time interval of tests and the number of consecutive negative results. The detection limits of the antigen test varied from $10^4$ copies/mL to $10^6$ copies/mL[14]. The threshold level for infectiousness is still uncertain and thus we investigated different values from $10^{4.5}$ copies/mL to $10^{5.5}$ copies/mL[6]. Simulations were separately performed for symptomatic patients and asymptomatic patients.

### Designing the isolation guideline utilizing antigen tests
In exploring different isolation guidelines, two metrics are considered: (1) the probability of prematurely ending isolation, and (2) the length of unnecessarily prolonged isolation, both of which are defined in the previous paper: "The probability of prematurely ending isolation is the chance that infected patients are released from isolation while they are still infectious. The length of prolonged isolation is defined as the difference between the time at which a patient is no longer infectious and the time when her or his isolation ends[6]". For simplicity, we define the first metric as "risk", and the second metric as "burden" of isolation. The risk is computed as the proportion of infected individuals with viral load above the infectiousness threshold values (and thus infectious) when the isolation ends. Specifically, denoting as $s_k$ the time when isolation of patient $k$ ends, the risk is calculated as $\sum_k I(V(s_k) < infectiousness\ threshold)/1000$, where $I$ is the identity function. The burden is computed as the mean difference between the time when isolation ends and the time when the viral load reaches the infectiousness threshold: $\sum_k (s_k - \hat{s}_k)/1000$, where $V(\hat{s}_k) = infectiousness\ threshold$. Note that the burden could take negative value, especially when less strict guidelines are implemented. We run 100 simulations (each simulation is composed of 1000 patients) and the mean and 95% prediction intervals of the distributions of the risk and burden are reported.

Balancing those two metrics is challenging because stricter guidelines (i.e., more consecutive negative results and longer intervals of tests) contribute to reducing the risk, however, yield unnecessarily long isolation. Therefore, the best guideline should be defined as the combination of the time interval of tests and the number of consecutive negative results which controls the risk of ending isolation of infectious patients under a certain level (1% or 5%) while minimizing the prolonged isolation.

The viral load data and codes that support the findings of this study are available at the repository, Zenodo[43].

### Reporting summary

Further information on research design is available in the Nature Research Reporting Summary linked to this article.

## Data availability

The viral load data that support the findings of this study are available at the repository, Zenodo[43].

## Code availability

All analyses were performed with the statistical computing software R (version 4.0.1). The analysis using nonlinear mixed effects model (including the algorithms for parameter estimation, such as Stochastic Approximation Expectation Maximization and Markov Chain Monte Carlo) was performed on MONOLIX 2019R2 (www.lixoft.com). Our code is publicly available at the repository, Zenodo[43].

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

## Acknowledgements

We would like to thank Yoshihiro Okada, Junichi Katada, Atsuhiko Wada, Kaku Irisawa, and Toshiki Takei at FUJIFILM Corporation for useful discussion. This study was supported in part by The Tokyo Foundation for Policy Research (to K.E. and K.S.); Grants-in-Aid for JSPS Scientific Research (KAKENHI) Scientific Research B 18KT0018 (to S.I), 18H01139 (to S.I.), 16H04845 (to S.I.), Scientific Research JP20H05921 (to K.A.), Scientific Research in Innovative Areas 20H05042 (to S.I.), 19H04839 (to S.I.), 18H05103 (to S.I.); AMED JP21dm0307009 (to K.A.); AMED CREST 19gm1310002 (to S.I.); AMED Japan Program for Infectious Diseases Research and Infrastructure, 20wm0325007h0001, 20wm0325004s0201, 20wm0325012s0301, 20wm0325015s0301 (to S.I.); AMED Research Program on HIV/AIDS 19fk0410023s0101 (to S.I.); AMED Research Program on Emerging and Re-emerging Infectious Diseases 19fk0108156h0001, 20fk0108140s0801 and 20fk0108413s0301 (to S.I.); AMED Program for Basic and Clinical Research on Hepatitis 19fk0210036h0502 (to S.I.); AMED Program on the Innovative Development and the Application of New Drugs for Hepatitis B 19fk0310114h0103 (to S.I.); Moonshot R&D Grant Number JPMJMS2021 (to K.A. and S.I.) and JPMJMS2025 (to S.I.); JST MIRAI program (JPMJMI22G1) (to S.I.); Mitsui Life Social Welfare Foundation (to S.I.); Shin-Nihon of Advanced Medical Research (to S.I.); Suzuken Memorial Foundation (to S.I.); Life Science Foundation of Japan (to S.I.); SECOM Science and Technology Foundation (to S.I.); The Japan Prize Foundation (to S.I.); Foundation for the Fusion of Science and Technology (to S.I.); Foundation of Kinoshita Memorial Enterprise (to S.I.); National Research Foundation of Korea (NRF) grant funded by the Korea government(MSIT) (2022R1C1C2003637) (to K.S.K.); Institute of AI and Beyond at the University of Tokyo (to K.A.); the MIDAS Coordination Center (MIDASSUGP2020-6) by a grant from the National Institute of General Medical Science (3U24GM132013-02S2) (to K.E. and M.A.). The study does not necessarily represent the views of the funding agencies listed above.

## Author contributions

Conceived and designed the study: K.E., S.I. Analyzed the data: Y.D.J., K.E., K.S.K., S.I. Wrote the paper: Y.D.J., K.E., K.S.K., W.J., S.I., Y.F., I.H.J., K.A., T.M., T.W., K.S., S.I., A.I.B., M.A. A.I.B. and M.A. are equally contributed last authors. All authors read and approved the final manuscript.

## Competing interests
The authors declare the following competing interests: M.A. has received research funding from Seqirus; the funding is not related to COVID-19. The other authors declare no competing interests.
