## [Peer Review File · Nature Communications]

Reviewers' Comments:

Reviewer #1:

Remarks to the Author:

In the submitted manuscript, the authors describe mathematical studies to evaluate rapid testing strategies for minimizing the risk of ending isolation measures in SARS-CoV-2 infected individuals. This topic is fundamentally of great relevance to fact-based decisions in public health response. Further scientific arguments are therefore very welcome.

The study is generally soundly described. Both the methodology used and the results and evaluations are comprehensible. In the discussion, the authors also already address some limitations of their study.

However, this paper would benefit from a significant addition: In the description of the methodology used, the authors explain that most of the primary studies used for the evaluation reported Ct values rather than absolute virus amounts. To compensate for this, a mathematical conversion of Ct values to virus quantities was performed.

However, such a conversion is actually not possible in such a simple way due to the methodological differences of the different PCR laboratory procedures. PCR assays can have very different linearities and detection limits. These then also affect the correlation of Ct values to absolute virus amounts. In laboratory assays, this can be compensated for by including quantitative standards, but this was not done (with the exception of source 19) in any of the primary studies used here.

The authors' calculations are therefore subject to at least one further methodological uncertainty. This should be taken into account in a revision of this manuscript, which is otherwise very well done.

Reviewer #3:

Remarks to the Author:

The authors present a mathematical modelling study, focussing on the impact of different isolation guidelines using antigen testing on both (i) risk of early release from isolation and (ii) the number of days spent unnecessarily isolating (i.e. below a certain threshold of viral load), which develops upon a previous study focussing upon PCR testing. The authors conclude that the risk of early release and burden of unnecessary isolation depends both upon the sensitivity of the antigen test, the threshold for which infectiousness is defined, and the number of consecutive test results required.

With antigen tests now forming part of isolation guidelines in countries such as the UK, such a piece of research is timely, and to my knowledge, reasonably novel (though the authors should include a discussion of other recent relevant studies, e.g.

<https://www.medrxiv.org/content/10.1101/2021.12.23.21268326v1>). The authors should be commended for considering both the costs and the benefits of shorter isolation periods using antigen testing. My main concerns about the manuscript in its current state surround the description of the methods and model fitting process, the quantification and presentation of uncertainty in results, and whether heterogeneity in viral loads between individuals is captured appropriately. I elaborate upon my main comments below, before providing more minor comments.

Main comments

1. The model fitting process is unclear from the methods section, and is not described in more detail in the supplementary information section. Could the authors provide references to the specific model fitting procedure (SAEM and empirical Bayes method) used in the paper, or include in the supplement a step-by-step description of the model fitting process. Could the authors include information surrounding prior distributions assumed for inferred parameters, and information regarding the number of iterations the model fitting process is run for. Could the authors also explain how the time of infection for each individual was estimated during the fitting?
2. In the supplement, point estimates are given for three model parameters (γ , β , and δ). However, posterior distributions are referred to. Would the authors be able to provide plots

showing the posterior distribution of these parameters, and also include information surrounding the range of values within the table? Further, the point estimate nor the distribution of values of estimated values for the parameter σ^2 is not included - could the authors provide this?

3. I'm unsure how heterogeneity between individuals is captured with the model. My understanding is that the authors fit parameters (beta, gamma, and epsilon) regarding the mean rates, and that distributions of these mean rates are inferred through the fitting process that best explain the data. If these represent population average values, it does not follow that the viral trajectory of individuals can be approximated sampling from these distributions, for example if there is a significant proportion of individuals who are superspreaders. If this is the case, it may be more appropriate to (i) sample N parameter sets from the posterior, (ii) simulate 1000 individuals for each parameter set, and (iii) present credible intervals surrounding estimates of risk and burden. In any case, could the authors expand upon how their approach captures heterogeneity between individuals.

4. Figures 2-4 only visualise mean values of risk and burden, and do not show the distribution of values obtained from the simulation. For Figure 4, bar plots could be replaced by violin plots. I acknowledge that including uncertainty in results would be hard in Figures 2 and 3, however, it may be useful to include supplementary figures focussing on particular rows/columns of the heat maps, with prediction intervals plotted explicitly.

5. A number of questions are raised by Figures 2 and 3:

(a) Firstly, why do the colour bars for 'length of unnecessary isolation' extend to a range below 0?

(b) In Figure 2, what is the explanation of the odd results regarding the probability of prematurely ending isolation - why in column A is a 4-day interval so much better than both longer and shorter intervals when 2 consecutive test results are needed (Similarly in column C, why is a 3-day interval so much better?)

6. The authors conclude that antigen tests "should be designed to have lower detection limits than infectiousness threshold value" in order to reduce the burden of unnecessary isolation. This result needs to be qualified, as tests with lower detection limits will result in higher risk of early release from isolation.

7. The authors state that longitudinal viral load data of symptomatic and asymptomatic COVID-19 patients were obtained from searches through PubMed and through Google Scholar - was this done in a systematic way? If so, could the authors provide information about the key word searches and refinement process used to identify relevant data. This would be particularly helpful for researchers interested in applying your method to more recent data, e.g. those interested in extending your method to data regarding vaccination.

8. Confidence intervals for risk and burden are calculated in different ways. For consistency, it may be better to use 2.5 and 97.5 percentiles of both, and refer to these as prediction intervals rather than confidence intervals. Moreover, the approximation used for the confidence intervals of the binomial distribution can be unreliable for small values of p, and a quick check of the stated risk ($p = 0.02$) does not result in the stated confidence interval (which I calculate as 0.0113-0.0287).

Suggestions/minor comments:

1. Could the authors upload the code to a public repository (e.g. Github?)

2. Is it possible to quantify the 'infectiousness' of those prematurely leaving isolation, using a metric other than the proportion who are still above the threshold (e.g. the number of days they remain infectious after isolation)?

3. Could the ordering of Figure 2 and 3 columns be changed? Such that the infectiousness threshold increases from left to right as in Figure 4?

4. Is it appropriate to model the error term as a random variable? Some individuals may consistently take swabs incorrectly, and therefore may consistently obtain false negative results.

5. The authors state that the isolation and the first test were performed 8 days after infection. However, some countries (like the UK) have used antigen tests to release individuals from isolation at an earlier stage (after five full days). Could the model be extended to include such instances? If not, would the authors be able to comment upon the potential implications of earlier release, and whether their model could be extended to investigate such scenarios.

6. Could the authors provide a discussion of other relevant studies (e.g. <https://www.medrxiv.org/content/10.1101/2021.12.23.21268326v1>)

7. Are there differences in viral trajectories between variants and wild-type patients from the data? While more recent variants (delta, and omicron) are not included in the model, would the authors be able to discuss the potential implications of new variants on their results?

8. The authors state “there are no publicly available data to calibrate the model for vaccinated individuals, regardless of vaccine types and numbers of doses” - is this still true? As stated in main point 6, a detailed description of the search method would be useful for future researchers wanting to replicate your method using new data.
9. Typo line 93 - 'laod' should be 'load'
10. In the abstract I would replace precociously with prematurely

Reviewers' comments:

Reviewer #1

In the submitted manuscript, the authors describe mathematical studies to evaluate rapid testing strategies for minimizing the risk of ending isolation measures in SARS-CoV-2 infected individuals. This topic is fundamentally of great relevance to fact-based decisions in public health response. Further scientific arguments are therefore very welcome. The study is generally soundly described. Both the methodology used and the results and evaluations are comprehensible. In the discussion, the authors also already address some limitations of their study. However, this paper would benefit from a significant addition: In the description of the methodology used, the authors explain that most of the primary studies used for the evaluation reported Ct values rather than absolute virus amounts. To compensate for this, a mathematical conversion of Ct values to virus quantities was performed. However, such a conversion is actually not possible in such a simple way due to the methodological differences of the different PCR laboratory procedures. PCR assays can have very different linearities and detection limits. These then also affect the correlation of Ct values to absolute virus amounts. In laboratory assays, this can be compensated for by including quantitative standards, but this was not done (with the exception of source 19) in any of the primary studies used here. The authors' calculations are therefore subject to at least one further methodological uncertainty. This should be taken into account in a revision of this manuscript, which is otherwise very well done.

***Response 1:** We would like to thank the reviewer for taking the time to carefully assess our study and for the positive evaluation.*

*We fully agree with the reviewer that the conversion between Ct values and viral load has limitations. As the reviewer pointed out, we used a simple conversion formula to calculate viral load (copies/ml) from Ct values (**Table 1**). This limitation is now acknowledged in the Discussion:*

“Third, we applied the same formula to convert Ct values to viral load despite this conversion should be determined for each study individually by considering the specific PCR assays that were used. However, such information was not available to us to carry out better conversions. To mitigate this limitation, we considered alternative equations to convert Ct values to viral loads and found consistent results.” (Page 10, Line 223 to 227)

*In addition, to assess the robustness of our results, we have now conducted a new sensitivity analysis following the approach used by Han et al., 2020, which is based on two different conversion formulas. The obtained results are similar to those obtained in the primary analysis (**Supplementary Figures 2 and 4**). The following paragraphs have been added to comment on these results:*

*“We performed several sensitivity analyses. First, we followed the approach of Han et al.27 and considered two alternative equations to convert Ct values to viral load. Although the obtained dynamics were slightly different depending on the selected conversion formula (**Supplementary Figure 2**), the overall impact on our main results was not substantial (**Supplementary Figure 4**).” (Page 7, Line 145 to 148)*

Reviewer #3

(General Comments)

The authors present a mathematical modelling study, focusing on the impact of different isolation guidelines using antigen testing on both (i) risk of early release from isolation and (ii) the number of days spent unnecessarily isolating (i.e. below a certain threshold of viral load),

which develops upon a previous study focusing upon PCR testing. The authors conclude that the risk of early release and burden of unnecessary isolation depends both upon the sensitivity of the antigen test, the threshold for which infectiousness is defined, and the number of consecutive test results required.

With antigen tests now forming part of isolation guidelines in countries such as the UK, such a piece of research is timely, and to my knowledge, reasonably novel (though the authors should include a discussion of other recent relevant studies, e.g. <https://www.medrxiv.org/content/10.1101/2021.12.23.21268326v1>). The authors should be commended for considering both the costs and the benefits of shorter isolation periods using antigen testing. My main concerns about the manuscript in its current state surround the description of the methods and model fitting process, the quantification and presentation of uncertainty in results, and whether heterogeneity in viral loads between individuals is captured appropriately. I elaborate upon my main comments below, before providing more minor comments.

Response 2: *We would like to thank the reviewer for taking the time to review our manuscript and the many useful suggestions. We also appreciate the very encouraging assessment made by the reviewer. As detailed in the responses below, we have deeply rewritten the manuscript to improve the description of the methodology and improve the presentation of the obtained results. We would also like to thank the reviewer for pointing us to the relevant manuscript by Bays and colleagues, which is now cited in our study.*

(Main Comments)

1. The model fitting process is unclear from the methods section, and is not described in more detail in the supplementary information section. Could the authors provide references to the specific model fitting procedure (SAEM and empirical Bayes method) used in the paper, or include in the supplement a step-by-step description of the model fitting process. Could the authors include information surrounding prior distributions assumed for inferred parameters, and information regarding the number of iterations the model fitting process is run for. Could the authors also explain how the time of infection for each individual was estimated during the fitting?

Response 3: *We apologize for the lack of detail. Briefly, to calibrate the viral dynamics model, we used nonlinear mixed-effect model accounting for fixed effect and random effect parameters. The fixed effect parameters were estimated using Stochastic Approximation Expectation Maximization; individual parameters (i.e., fixed + random effect) were subsequently estimated using Markov Chain Monte Carlo. The detailed description of model fitting procedure has been added to the Supplementary Information (See **Supplementary Note 1**).*

2. In the supplement, point estimates are given for three model parameters (gamma, beta, and delta). However, posterior distributions are referred to. Would the authors be able to provide plots showing the posterior distribution of these parameters, and also include information surrounding the range of values within the table? Further, the point estimate nor the distribution of values of estimated values for the parameter σ^2 is not included - could the authors provide this?

Response 4: *We apologize for the omission. We have updated **Supplementary Table 1**, which now reports the estimated 95% CI of the model parameters and variance of the measurement error σ^2 . As suggested by the reviewer, we have added **Supplementary Figure 1** showing the*

posterior distributions of the fixed effect parameters and the distribution of the measurement error.

3. I'm unsure how heterogeneity between individuals is captured with the model. My understanding is that the authors fit parameters (beta, gamma, and epsilon) regarding the mean rates, and that distributions of these mean rates are inferred through the fitting process that best explain the data. If these represent population average values, it does not follow that the viral trajectory of individuals can be approximated sampling from these distributions, for example if there is a significant proportion of individuals who are superspreaders. If this is the case, it may be more appropriate to (i) sample N parameter sets from the posterior, (ii) simulate 1000 individuals for each parameter set, and (iii) present credible intervals surrounding estimates of risk and burden. In any case, could the authors expand upon how their approach captures heterogeneity between individuals.

Response 5: *We apologize for the lack of detail. To estimate the distributions of model parameters we considered the sum of fixed effect (i.e., same for all individuals) and random effect (i.e., individual variability). In the simulations, we then sample values of the parameters from the estimated joint distribution. We have added **Supplementary Figure 8** (which is appended below for reviewer convenience) to show the individual variability captured by the model and thus reflected in our simulations. Note that, as we analyzed 210 patients in total, we randomly selected 60 symptomatic and asymptomatic individuals from different locations to be*

included in the figure.

Moreover, as suggested by the reviewer, all new analyses are based on 100 simulations, each composed by 1,000 patients (See **Response 10**). We updated the description of the simulation as follows:

“Parameter values of the simulation for each patient were resampled from the posterior joint distributions of individual parameters estimated in the fitting process (Supplementary Figure 8).” (Page 13, Line 278 to 279).

and

“We run 100 simulations (each simulation is composed of 1,000 patients) and the mean and 95% prediction intervals of the distributions of the risk and burden are reported.” (Page 14, Line 301 to 303).

4. Figures 2-4 only visualise mean values of risk and burden, and do not show the distribution of values obtained from the simulation. For Figure 4, bar plots could be replaced by violin plots. I acknowledge that including uncertainty in results would be hard in Figures 2 and 3, however, it may be useful to include supplementary figures focusing on particular rows/columns of the heat maps, with prediction intervals plotted explicitly.

Response 6: *Thank you for the excellent suggestions. We have added **Supplementary Figure 3**, showing the distributions of the risk and burden for several representative scenarios. Moreover, **Figure 4** now uses by violin plots.*

5. A number of questions are raised by Figures 2 and 3:

(a) Firstly, why do the colour bars for ‘length of unnecessary isolation’ extend to a range below 0?

(b) In Figure 2, what is the explanation of the odd results regarding the probability of prematurely ending isolation - why in column A is a 4-day interval so much better than both longer and shorter intervals when 2 consecutive test results are needed (Similarly in column C, why is a 3-day interval so much better?)

Response 7: *We apologize for the lack of clarify.*

(a) The definition of “length of unnecessarily prolonged isolation” (burden) is the difference between the time when isolation ends and the time when the viral load reaches the infectiousness threshold; therefore, negative values are possible. We carefully defined both the risk and burden and mentioned that the burden could take negative values:

“The risk is computed as the proportion of infected individuals with viral load above the infectiousness threshold values (and thus infectious) when the isolation ends. Specifically, denoting as s_k the time when isolation of patient k ends, the risk is calculated as $\sum_k I(V(s_k) < \text{infectiousness threshold})/1000$, where I is the identity function. The burden is computed as the mean difference between the time when isolation ends and the time when the viral load reaches the infectiousness threshold: $\sum_k (s_k - \hat{s}_k)/1000$, where $V(\hat{s}_k) = \text{infectiousness threshold}$. Note that the burden could take negative value, especially when less strict guidelines are implemented.” (Page 14, Line 295 to 301)

(b) We are grateful for the reviewer for pointing out this result, which was caused by the stochasticity of the simulations. We have now re-run the experiments using a larger number of simulations (100 simulations, each with 1,000 patients – as suggested by the reviewer in a previous comment) and now all the results are as expected.

6. The authors conclude that antigen tests “should be designed to have lower detection limits than infectiousness threshold value” in order to reduce the burden of unnecessary isolation. This result needs to be qualified, as tests with lower detection limits will result in higher risk of early release from isolation.

Response 8: *We apologize for the lack of clarity. VL goes in the opposite direction than Ct. Therefore, one should be careful of possibly increasing false positive when detection limit (in VL) increases. We added the following comment to the main text to empathize that we used viral load rather than Ct values in this paper.*

“We stress that our analyses use viral load rather than Ct value. Therefore, the detection limit represents the lowest value of the viral load that an antigen test can detect. Similarly, the infectiousness threshold represents the lowest value of the viral load for which SARS-CoV-2 transmission may occur.” (Page 4, Line 81 to 84)

7. The authors state that longitudinal viral load data of symptomatic and asymptomatic COVID-19 patients were obtained from searches through PubMed and through Google Scholar - was this done in a systematic way? If so, could the authors provide information about the key word searches and refinement process used to identify relevant data. This would be particularly helpful for researchers interested in applying your method to more recent data, e.g. those interested in extending your method to data regarding vaccination.

Response 9: *Thank you for this very nice suggestion. We used the following query in PubMed to identify papers with longitudinal viral load data:*

("COVID-19" or "SARS-CoV-2") and ("viral load" or "viral loads" or "viral titer" or "cycle threshold" or "cycle thresholds" or "viral RNA concentration" or "viral RNA concentrations" or "viral shedding") and ("peak" or "kinetics" or "clinical course")

Papers published in 2020 and 2021 were included for further investigation. In total, 250 papers were identified. We reviewed each paper to extract the relevant data based on the following inclusion criteria: 1) viral load was measured at least at three different time points; 2) viral load was measured from upper respiratory specimens (i.e., nose or pharynx); 3) patients were not treated with antiviral drugs or vaccinated before infection (because the model does not account for vaccine and antiviral effect). The papers meeting these criteria were 7. Further, we identified 3 additional papers by our regular search of the literature through PubMed and Google Scholar. We added this information in the main text (Page 12, Line 237 to 251).

8. Confidence intervals for risk and burden are calculated in different ways. For consistency, it may be better to use 2.5 and 97.5 percentiles of both, and refer to these as prediction intervals rather than confidence intervals. Moreover, the approximation used for the confidence intervals of the binomial distribution can be unreliable for small values of p , and a quick check of the stated risk ($p = 0.02$) does not result in the stated confidence interval (which I calculate as 0.0113-0.0287).

Response 10: *Thank you for the careful reading. As suggested in a previous comment, we now run 100 simulations with 1,000 patients each; results are now reported together with 95% prediction intervals.*

(Suggestions / Minor Comments)

1. Could the authors upload the code to a public repository (e.g. Github?)

Response 11: As suggested, we have now uploaded our code to a public repository. The Data availability statement has been amended as follows:

“All analyses were performed with the statistical computing software R (version 4.0.1). The analysis using nonlinear mixed effects model was performed on MONOLIX 2019R2 (www.lixoft.com). Our code is publicly available at https://github.com/Yongdam-Jeong/SARS-CoV-2_Isolation_antigen_test.” (Page 15, Line 310 to 313).

2. Is it possible to quantify the ‘infectiousness’ of those prematurely leaving isolation, using a metric other than the proportion who are still above the threshold (e.g. the number of days they remain infectious after isolation)?

Response 12: Thank you for this very interesting suggestion. We have added a new figure (**Supplementary Figure 7**) based on the metric proposed by the reviewer. We have added the following paragraph to the main text to comment on the obtained results:

*“The proportion of infectious patients (i.e., risk) does not capture how long prematurely released individuals are still infectious. To capture it, we have added a new metric: the mean number of days an infected individual is infectious after the end of the isolation. The obtained results are reported in **Supplementary Figure 7**). For example, when an infectiousness threshold value of $10^{5.0}$ copies/mL, detection limit $10^{4.0}$ copies/mL, tests performed every day, and 2 consecutive negative results to end isolation of symptomatic individuals, the mean number of days was 2.1 days (95%PrI: 1 to 6).” (Page 7, Line 159 to 164)*

3. Could the ordering of Figure 2 and 3 columns be changed? Such that the infectiousness threshold increases from left to right as in Figure 4?

Response 13: Figures 2 and 3 were updated accordingly.

4. Is it appropriate to model the error term as a random variable? Some individuals may consistently take swabs incorrectly, and therefore may consistently obtain false negative results.

Response 14: We agree with the reviewer that this may be a possibility. Therefore, we have added a sensitivity analysis where we assumed that the error is consistent over time for each patient. This analysis leads to a higher estimation of the risk, which is associated with patients with consistently undermeasured VL. The obtained results are reported in **Supplementary Figure 6** and are presented in the main text:

*“Third, we considered a fixed measurement error of the viral load for each patient over the course of infection. The overall results are in agreement with the main analysis; however, we estimated a general increased risk of prematurely ending isolation, which is associated with patients with consistently undermeasured VL thus resulting in an earlier release (see **Supplementary Figure 6** as compared to **Figure 2**).” (Page 7, Line 154 to 158)*

5. The authors state that the isolation and the first test were performed 8 days after infection. However, some countries (like the UK) have used antigen tests to release individuals from isolation at an earlier stage (after five full days). Could the model be extended to include such instances? If not, would the authors be able to comment upon the potential implications of earlier release, and whether their model could be extended to investigate such scenarios.

Response 15: We are glad to add the analysis suggested by the reviewer where tests start 5 days after infection. The obtained results (which are reported in **Supplementary Figure 5**) are very consistent with those of the main analysis. The following paragraph has been added to the main text:

*“Second, we set the initial day of the testing to be 5 days after the infection event (as compared to 8 days used in the main analysis). The obtained results are very consistent with those obtained in the main analysis (see **Supplementary Figure 5** as compared to **Figure 2**). In fact, the viral load is much higher than the detection limit both at 5 and 8 days after infection; thus the false-negative rate is extremely low at both times (**Figure 1**).” (Page 7, Line 149 to 153)*

6. Could the authors provide a discussion of other relevant studies (e.g. <https://www.medrxiv.org/content/10.1101/2021.12.23.21268326v1>)

Response 16: Thank you for suggesting this relevant study. This, together with other studies, is cited and discussed in the main text:

“Other studies have assessed the impact of using antigen tests in the context of ending isolation. All these studies consistently concluded that using antigen test may reduce redundant isolations or prevent onward transmission. Similar to what we have done, Peng et al. and Quilty et al. considered the temporal change in the viral load, which in turns affect the transmission potential and test sensitivity. However, differently from those studies (which are based on piece-wise models or cubic Hermite splines), here we used a model that provides a biological explanation of the dynamics of viral load and can thus be refined to consider other factors shaping viral dynamics as, for instance, the use of an antiviral treatment.” (Page 10, Line 208 to 214)

7. Are there differences in viral trajectories between variants and wild-type patients from the data? While more recent variants (delta, and omicron) are not included in the model, would the authors be able to discuss the potential implications of new variants on their results?

Response 17: Unfortunately, we do not have enough sample size to feel comfortable in evaluating possible differences between SARS-CoV-2 variants. One consideration that we can make is that we found differences in the viral dynamics of symptomatic and asymptomatic infections. Specifically, the VL decay rate was faster in asymptomatic cases than symptomatic cases (although the 95%PrIs overlap), thus the viral load is expected to fluctuate around the detection limit for shorter period in asymptomatic cases, which led to lower risk and less burden under the same guideline. As infection from different variants appear to be associated with different severity rates (including probability of developing any symptom), if the same guideline to end the isolation period is applied both to symptomatic and asymptomatic infections, its effectiveness can be variant-specific. We have added the following paragraph to comment on this:

“Our study highlights differences in the viral dynamics of symptomatic and asymptomatic infections. Specifically, we found that the viral load decays quicker in asymptomatic patients than symptomatic patients (although the 95%PrIs overlap); thus, the viral load of asymptomatic individuals fluctuates around the detection limit for shorter period, which leads to lower risk and less burden if the same guideline is applied to both symptomatic and asymptomatic patients. As infection from different variants appear to be associated with different severity rates (including probability of developing any symptom), if the

same guideline to end the isolation period is applied both to symptomatic and asymptomatic infections, the overall effectiveness of that guideline can vary for different SARS-CoV-2 variants.” (Page 10, Line 200 to 207)

8. The authors state “there are no publicly available data to calibrate the model for vaccinated individuals, regardless of vaccine types and numbers of doses” - is this still true? As stated in main point 6, a detailed description of the search method would be useful for future researchers wanting to replicate your method using new data.

Response 18: *We apologize for the misleading sentence. At the time we conducted our literature review (December 2021 – see response to a previous comment), no studies were available. We have rephrased the sentence as follows: “Second, as of December 2021 (the date we conducted our literature review), there were no publicly available data to calibrate the model for vaccinated individuals, regardless of vaccine types and numbers of doses.” (Page 10, Line 220 to 221)*

9. Typo line 93 - 'laod' should be 'load'

Response 19: *Thank you - we fixed the typo.*

10. In the abstract I would replace precociously with prematurely

Response 20: *Thank you - correction made.*

Reviewers' Comments:

Reviewer #1:

Remarks to the Author:

I thank the authors for their thoughtful and careful consideration of the comments in the first review.

I have no further remarks and am convinced that this manuscript as it stands now is suitable for publication in Nature Communications.

Reviewer #3:

Remarks to the Author:

I thank the authors for their responses and the additional work undertaken to address my previous comments. However, I still have two central concerns regarding (1) the full presentation of results and uncertainty from the fitting process, and (2) whether the model structure adequately captures variability between individuals. I elaborate on each of these specific points in turn.

1. Important information regarding the fitting process is still omitted from the paper, and the underlying uncertainty from parameters is not always displayed in figures. Specifically:

a) The posterior distributions plotted (Figure S1) are smooth functions - if parameters are estimated through MCMC, I expected the distribution of posteriors to be presented as histograms, which will not be smooth owing to the inherent randomness of MCMC approaches. Moreover, the figure legend lacks necessary detail to interpret the figure - how many samples of the posterior are taken and plotted to obtain these figures?

b) Prior distributions were chosen from the authors' "experience on viral dynamics modelling". Could the authors please elaborate on their justification of their priors - is there any references in the literature that can be cited to support these choices? If not, and if posterior distributions are not influenced by the choice of prior distributions, I would suggest the use of more uninformative priors.

c) As τ_k is jointly estimated with the other parameters (stated in Supplementary Text 2), its prior should be stated in supplementary note 1, and should be stated as a set of parameters estimated within the main text. An example of its posterior for one individual could appear as a panel in supplementary Figure 1 or as its own figure. Information regarding the distribution of such values should also appear, either in Supplementary note 1 or incorporated in some way into supplementary table 1.

d) In some supplementary figures there is no attempt to visualise the parametric uncertainty within the model (Supplementary Figure 2 and Supplementary Figure 8). These could be plotted with regions depicting 95% prediction intervals.

e) All figures lack some relevant information for their interpretation, i.e. the number of parameter sets simulated and the number of individuals within each simulation.

2. I am still not convinced that the model captures individual variability in the way that it intends. The authors state in their response model accounts for fixed effect parameters (γ , β , and δ) which are the same for all individuals, and for a random effect (ϵ), which changes for each individual, corresponding to measurement error. This raises two questions:

a) Firstly, is it supposed that the error in measurement is the only source of variability between the viral loads of individuals? This seems an overly simplistic account. In any case, the assumptions underpinning the way in which individual variability is incorporated should be described in the methods of the paper (in plain english rather than in terms of parameters), while the limitations of this approach should be reflected upon in the discussion.

b) Secondly, the main text states that the variance of error measurement is estimated during the fitting process. However, the previous description seems to imply that an epsilon is fitted for each specific individual? Can this be clarified?

c) In supplementary table 1, only the point estimate of the variance of error measurement is given. If this is inferred through MCMC, there should be a distinct variance of error measurement corresponding to each parameter set. Accordingly, the 95% credible interval for this should also be displayed . If the variance of error measurement is fitted before the other parameters, how and why this is done needs to be outlined Supplementary Note 1 - if gamma, beta, and delta have different values, how does this not impact σ^2 ?

Reviewers' comments:

Reviewer #1

I thank the authors for their thoughtful and careful consideration of the comments in the first review. I have no further remarks and am convinced that this manuscript as it stands now is suitable for publication in Nature Communications.

***Response 1:** We are delighted to hear that the reviewer is now satisfied with our revision. We truly appreciate their time and effort in reviewing and improving our manuscript.*

Reviewer #3

I thank the authors for their responses and the additional work undertaken to address my previous comments. However, I still have two central concerns regarding (1) the full presentation of results and uncertainty from the fitting process, and (2) whether the model structure adequately captures variability between individuals. I elaborate on each of these specific points in turn.

Response 2:

We would like to thank the reviewer for further assessing our manuscript and the very constructive feedback that surely helped us to improve our study. The responses to each reviewer's comment are detailed below.

1. Important information regarding the fitting process is still omitted from the paper, and the underlying uncertainty from parameters is not always displayed in figures. Specifically:

a) The posterior distributions plotted (Figure S1) are smooth functions - if parameters are estimated through MCMC, I expected the distribution of posteriors to be presented as histograms, which will not be smooth owing to the inherent randomness of MCMC approaches. Moreover, the figure legend lacks necessary detail to interpret the figure - how many samples of the posterior are taken and plotted to obtain these figures?

***Response 3:** We fully agree with the reviewer that presenting a smoothed density distribution could mislead the reader. As suggested by the reviewer, we now show the histogram of each parameter distribution (see updated **Supplementary Figure 1**). In the figure caption, we added the information about the number of samples used to generate the histograms (specifically, 10,000 samples).*

b) Prior distributions were chosen from the authors' "experience on viral dynamics modelling". Could the authors please elaborate on their justification of their priors - is there any references in the literature that can be cited to support these choices? If not, and if posterior distributions are not influenced by the choice of prior distributions, I would suggest the use of more uninformative priors.

***Response 4:** We would like to thank the reviewer for this very constructive comment. First, we apologize of the lack of clarity. The prior distributions used in the main analysis correspond to the posterior distribution of a previous paper modeling the viral dynamics of SARS-CoV-2 (albeit using different data and not differentiating between asymptomatic and symptomatic patients). The sentence has been revised as follows:*

“To guarantee the positiveness of the parameters (i.e., negative values do not biologically make sense), lognormal distributions were used as prior distributions. These prior distributions correspond to the posterior distributions obtained in a previous study³.” (Page 13 in Supplementary Information)

*Second, we have re-run the MCMC procedure using uninformative prior distributions. The obtained results are very consistent with those obtained assuming informative priors. The estimated posterior distributions are presented in **Supplementary Figures 9** and the values of the estimated parameters are compared in **Supplementary Table 2**. We have also added the following sentence to comment on these findings:*

*“To guarantee the reliability of our results, we also performed a sensitivity analysis where we used uninformative prior distributions. As shown in **Supplementary Figure 9** and **Supplementary Table 2**, the obtained results are robust to changes of the prior distributions.” (Page 13-14 in Supplementary Information)*

c) As τ_k is jointly estimated with the other parameters (stated in Supplementary Text 2), its prior should be stated in supplementary note 1, and should be stated as a set of parameters estimated within the main text. An example of its posterior for one individual could appear as a panel in supplementary Figure 1 or as its own figure. Information regarding the distribution of such values should also appear, either in Supplementary note 1 or incorporated in some way into supplementary table 1.

Response 5: *Thank you for this suggestion, which, we believe, has highly improved the clarity of our study. The description of the parameter τ has now been added to the main text:*

“The time origin of the longitudinal viral load data corresponds to the time after symptom onset (for symptomatic patients) and the time after diagnosis (for asymptomatic patients). Therefore, we estimated a further model parameter, τ , which represents the time interval between infection to symptom onset for symptomatic patients or to diagnosis for asymptomatic patients (see Supplementary Note 1 for detail).” (Page 13, Line 260 to 264).

*Further, the prior distribution of τ has been added to **Supplementary Note 1** with the other parameters (See **Response 4** regarding prior distributions). Furthermore, its posterior distribution has been added to **Supplementary Figure 1**, and the median and 95%CI has been added to **Supplementary Table 1**.*

d) In some supplementary figures there is no attempt to visualise the parametric uncertainty within the model (Supplementary Figure 2 and Supplementary Figure 8). These could be plotted with regions depicting 95% prediction intervals.

Response 6: *We apologize for this omission. The 95% predictive intervals have now been added to **Supplementary Figures 2 and 8**.*

e) All figures lack some relevant information for their interpretation, i.e. the number of parameter sets simulated and the number of individuals within each simulation.

Response 7: *We apologize for this lack of detail. Figure captions have been updated to include the number of simulations and patients in each simulation.*

2. I am still not convinced that the model captures individual variability in the way that it intends. The authors state in their response model accounts for fixed effect parameters (gamma, beta, and delta) which are the same for all individuals, and for a random effect (epsilon), which changes for each individual, corresponding to measurement error. This raises two questions:

a) Firstly, is it supposed that the error in measurement is the only source of variability between the viral loads of individuals? This seems an overly simplistic account. In any case, the assumptions underpinning the way in which individual variability is incorporated should be described in the methods of the paper (in plain english rather than in terms of parameters), while the limitations of this approach should be reflected upon in the discussion.

b) Secondly, the main text states that the variance of error measurement is estimated during the fitting process. However, the previous description seems to imply that an epsilon is fitted for each specific individual? Can this be clarified?

c) In supplementary table 1, only the point estimate of the variance of error measurement is given. If this is inferred through MCMC, there should be a distinct variance of error measurement corresponding to each parameter set. Accordingly, the 95% credible interval for this should also be displayed. If the variance of error measurement is fitted before the other parameters, how and why this is done needs to be outlined Supplementary Note 1 - if gamma, beta, and delta have different values, how does this not impact σ^2 ?

Response 8:

We apologize for the lack of clarity. In light of the reviewer's comment, we have realized that the term "measurement error" was rather misleading. In the revised version, we have renamed it simply as "error". We defined the error as the difference between the observed viral load and the viral load estimated by the model. We believe that using this new term helps clarifying that the error is not simply due to errors in the empirical measurement of the viral load for each patient, but it can be caused by various reasons that are not captured by the model.

Second, in the fitting process, we used a nonlinear mixed effect model, which considers both a fixed effect and a random effect. The random effect represents the individual variability (which is distinct from the error).

*The definition of the error and how its variance is estimated has been added to the **Supplementary Note 1** ("Estimation of the variance of the error"). Moreover, the following sentence has been added to the main text to clarify this point:*

*"The "true" viral load data, $V(t)$, for 1,000 simulated patients was estimated by running the developed viral dynamics model. Parameter values of the simulation for each patient were sampled from the joint posterior distributions of model parameters (as estimated in the fitting process **Supplementary Figure 8**). The measured viral load is assumed as a sum of the true viral load and the error: $\hat{V}(t) = V(t) + \varepsilon$, $\varepsilon \sim N(0, \sigma)$, where ε is the error term, which is defined as the difference between the viral load reported in the data and the viral load estimated by the calibrated model. The variance of the error term, σ^2 , was estimated in the fitting process (see **Supplementary Note 1**)."* (Page 13-14, Line 277 to 283)

We would like to thank the reviewer once again for this comment that gave us the chance to better clarify this important aspect of our methodology.

Reviewers' Comments:

Reviewer #3:

Remarks to the Author:

I would like to thank the authors for their diligent revisions and responses. Each of the points raised in my previous review have been adequately addressed. I have only a few minor suggestions:

1. On line 125 of the supplementary information, I would suggest using the phrase 'To assess the reliability of our results' rather than 'to guarantee the reliability of our results.'
2. Can the uninformative priors used for Supplementary Figure 9 be specified (possibly just in the figure caption)? These appear to be different for each parameter.
3. Rapid antigen tests are often referred to as lateral flow tests. I would suggest including this terminology in brackets at some point in your introduction.

Reviewers' comments:

Reviewer #3 (Remarks to the Author):

I would like to thank the authors for their diligent revisions and responses. Each of the points raised in my previous review have been adequately addressed. I have only a few minor suggestions:

Response 1: We are delighted to hear that the reviewer is satisfied with our revision. We truly appreciate their time and effort in reviewing and improving our manuscript.

1. On line 125 of the supplementary information, I would suggest using the phrase 'To assess the reliability of our results' rather than 'to guarantee the reliability of our results.'

Response 2: We updated the sentence accordingly.

2. Can the uninformative priors used for Supplementary Figure 9 be specified (possibly just in the figure caption)? These appear to be different for each parameter.

Response 3: We added the information of the prior distributions in the figure caption.

3. Rapid antigen tests are often referred to as lateral flow tests. I would suggest including this terminology in brackets at some point in your introduction.

Response 4: We added the term "lateral flow tests" in the introduction section.